# Representation Learning for Object Detection from Unlabeled Point Cloud Sequences

**Xiangru Huang**
MIT CSAIL
xrhuang@csail.mit.edu

**Yue Wang**[*]
NVIDIA Research & MIT CSAIL
yuewang@csail.mit.edu

**Vitor Guizilini**
Toyota Research Institute (TRI)
vitor.guizilini@tri.global

**Rares Ambrus**
Toyota Research Institute (TRI)
rares.ambrus@tri.global

**Adrien Gaidon**
Toyota Research Institute (TRI)
adrien.gaidon@tri.global

**Justin Solomon**
MIT CSAIL
jsolomon@mit.edu

**Abstract:** Although unlabeled 3D data is easy to collect, state-of-the-art machine learning techniques for 3D object detection still rely on difficult-to-obtain manual annotations. To reduce dependence on the expensive and error-prone process of manual labeling, we propose a technique for representation learning from unlabeled LiDAR point cloud sequences. Our key insight is that moving objects can be reliably detected from point cloud sequences without the need for human-labeled 3D bounding boxes. In a single LiDAR frame extracted from a sequence, the set of moving objects provides sufficient supervision for single-frame object detection. By designing appropriate pretext tasks, we learn point cloud features that generalize to both moving and static unseen objects. We apply these features to object detection, achieving strong performance on self-supervised representation learning and unsupervised object detection tasks. Code is available at https://github.com/xiangruhuang/PCSeqLearning

**Keywords:** Representation learning, object detection, point cloud sequences

Among the modalities used for object detection in autonomous driving, LiDAR point clouds capture accurate 3D scene structure, yielding state-of-the-art performance [1, 2]. Yet, sparsity and irregularity make it difficult for models to generalize to complicated real-world environments. Moreover, object detection requires several tasks to be solved jointly, including foreground–background segmentation, instance segmentation, object localization, and classification. This results in a high demand for human labels of object locations, velocities, orientations, and other properties.

Unlike expensive human-generated labels, *unlabeled* point cloud sequences can easily be collected by autonomous vehicles with LiDAR sensors whenever they are on the road. These temporally-ordered sequences contain more information than individual frames. For example, we can estimate correspondences between adjacent point clouds. Recent works distill information learned from point cloud sequences to train single-frame models for motion estimation, object detection, prediction, and motion tracking [3, 4, 5, 6, 7, 8, 9, 10, 11]. These works typically focus on a sub-module relevant to object detection (e.g., scene flow), or require many labeled sequences to promote generalization.

We propose a representation learning approach to learning features for object detection from unlabeled LiDAR point cloud sequences. Unlike previous works, ours is capable of learning features without 3D bounding box annotations, with competitive performance on object detection benchmarks among unsupervised and self-supervised approaches. To deliver generalization from limited labeled data, we use geometry processing techniques to derive a pseudo-label generator. This generator ingests unlabeled point cloud sequences and produces annotations for pretext tasks like motion segmentation and moving object detection. Then, we use the generated annotations to pretrain a *single-frame* feature extractor that can be used for downstream tasks like object detection.

Our pseudo-label generator identifies moving objects as sequences of point clusters across frames (Figure 2). Unlike existing single-frame LiDAR segmentation methods that focus on point connectivity [12, 13, 14], we estimate motion and planar structures from each sequence. This information

---

[*]Work done while at MIT.

6th Conference on Robot Learning (CoRL 2022), Auckland, New Zealand.

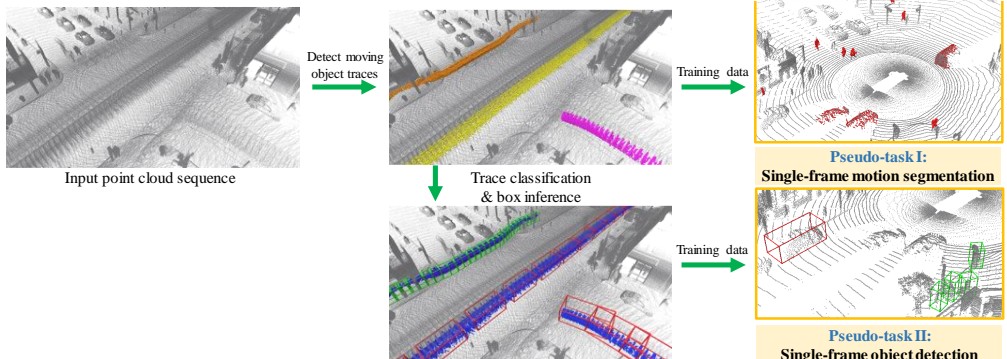

Figure 1: Overview of our approach. Detected moving objected from sequences are fed into later self-supervised tasks.

enables us to separate out moving objects reliably, yielding highly-accurate moving object detection. By combining all detected point clusters in the sequence, we further estimate the location, orientation, and category of each moving object. To promote generalization and mitigate the bias of working exclusively with moving objects, we propagate estimated pseudo-labels to static objects following a variant of [11]. The key difference of our approach is that we check the consistency of predicted object labels across frames to prune incorrect predictions, improving robustness of label propagation.

To summarize, our contributions are as follows:

- An algorithm for the detection of moving objects from unlabeled point cloud sequences, which enables self-supervised representation learning for point cloud data.
- Two representation learning pseudo-tasks leveraging our detected moving objects for the downstream task of object detection, without requiring manually-annotated labels at any stage of training.
- Competitive self-supervised object detection accuracy and state-of-the-art unsupervised object discovery accuracy.

# 1 Related Work

**Point cloud self-supervised learning.** Self-supervision is popular for representation learning from unlabeled data. For point clouds, several self-supervised pretext tasks have been proposed. Sauder and Sievers [15] split point clouds into voxels and reassemble shuffled voxels. Zhang et al. [16] enforce consistency between point cloud and voxel representations. Xie et al. [17] synthesize transformed point clouds and enforce consistency. P4Contrast uses point-pixel pairs to learn complementary features from point clouds and images [18]. Liu et al. [19] include negative pairs in P4Contrast. Tian et al. [20] learn to discover unseen objects from image and LiDAR clues. Our work is closely related to representation learning and unsupervised object discovery method, using relationships between adjacent frames to augment available information.

**3D object detection.** 3D object detection works can be categorized by data type, e.g. voxels [21], points [22, 23], range images [24, 25], and hybrid representations [26, 27, 28]. Many works incorporate temporally-adjacent frames to improve detection. Luo et al. [29] stack frames into a sparse 4D tensor, using 3D convolutions for efficiency. Hu et al. [30] incorporate temporal voxel occupancy estimated from raycasting. Several models deal with spatio-temporal information, such as Transformers [31] and LSTMs [32]. Ku et al. [33], Chen et al. [34], Xu et al. [35] combine modalities like point clouds and images into a single framework. Frustum-PointNet leverages 2D object detectors to form a frustum crop of points and then uses PointNet to aggregate features [36]. Beyond visual input, Yang et al. [37] use high-definition maps to boost performance of 3D object detectors. Liang et al. [38] argue that multi-tasking improves representations over single-tasking. Our method is also closely related to [39], which learns a perceptual model for unknown classes.

**Learning from point cloud sequences.** Point cloud sequence datasets have grown in scale and diversity, covering tasks like 3D reconstruction [40, 41], semantic segmentation [42], and object detection [1, 2, 43]. Although sequences often yield better performance than single frames, they require new representations. Early works treated them as 4D grids [44]. MeteorNet includes spatio-temporal neighborhoods for dynamic point cloud sequences [45]. CaSPR normalizes the temporal dimension [9], yielding a time-continuous representation. Mersch et al. [5] convert point sequences

|                          | Car     | Pedestrian | Cyclist |
|--------------------------|---------|------------|---------|
| Box Count                | 4352210 | 2037627    | 49518   |
| Moving Box Count         | 1161019 | 1427211    | 42411   |
| Moving Trace Count       | 12496   | 13052      | 438     |
| Velocity in $[0.05, 2]$ m/s | 850640  | 1044324    | 37051   |
| Turning Angle $< 3°$     | 1153865 | 1379915    | 41546   |

Table 1: Waymo Open training dataset statistics [1] (798 sequences), including the number of (moving or all) bounding boxes, the number of unique moving objects and the velocity and turning angle distribution of all moving bounding boxes. We focus on the fraction of moving boxes that follow a smooth trajectory.

into stacked 2D range images, processed using 3D convolution. Qi et al. [11] use motion and complementary views to help detect moving and static objects. Compared to previous work, our approach performs object detection without requiring annotated 3D bounding boxes.

## 2 Approach Overview

Our framework takes two main steps, summarized in Figure 1: (1) extract a set of moving objects (defined below as *object traces*) from a temporally-ordered sequence of unlabeled point clouds; and (2) train a feature extraction module using these object traces. We do *not* require human-labeled 3D bounding boxes at any stage. Here, we introduce the concept of object traces and discuss why they are necessary for our approach. We then elaborate the key observations and challenges in each step.

**Object traces.** Point cloud segmentation generally involves finding instances of objects, represented by sets of point clusters $\{C_i\}$ where $C_i \in \mathbb{R}^{M_i \times 3}$. In point cloud sequences, object instances appear consistently in consecutive frames. Therefore, we are interested in *object traces*, which are sequences of point clusters across consecutive frames corresponding to the same object.

Formally, given a point cloud sequence $\mathcal{P} = \{P_1, \ldots, P_N\}$ where $P_t$ is the $t$-th frame from a LiDAR camera, an *object trace* is a temporally-ordered sequence of clusters $\mathcal{C} = \{C_l, \ldots, C_r\}$, where $C_t \subseteq P_t$ for any $1 \le l \le t \le r \le N$. Figure 2 shows an example.

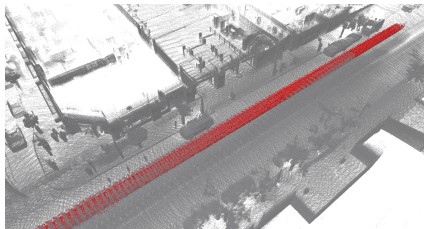

Our key observation is that many object classes relevant for autonomous driving are *dynamic*, including pedestrians, vehicles, and cyclists. Statistics from the Waymo dataset show that many ground truth object traces follow smooth trajectories (Table 1). Intuitively, the length and motion consistency of object traces reduce uncertainty in their detection, enabling our robust non-learning trace detection algorithms from unlabeled sequences. Detected traces can be used in subsequent pseudo-tasks.

Figure 2: An *object trace* is a sequence of point clusters moving along a smooth trajectory. Here, we highlight an object trace of a moving vehicle.

**Object trace detection.** Detecting object traces is inevitably disturbed by interactions between objects and the environment and irregular motion and sampling density across frames. Even supervised methods suffer from detection uncertainty. Here, we focus on objects moving along smooth trajectories. Our detector optimizes for detection precision instead of coverage. This choice only adds a mild bias to the detected objects since velocities are independent from geometric appearance.

Our object trace detection follows a standard proposal-and-rejection framework. For each sequence, we propose candidate clusters corresponding to movable objects. Then, Kalman filtering acquires object traces according to their motion. We collect a subset of object traces with smooth trajectories.

**Self-supervision.** Given our high-quality detected object traces from unlabeled point cloud sequences, we design two self-supervised tasks that enable representation learning for point clouds.

The rest of our paper is organized as follows. §3 details our object trace detection algorithm. §4 introduces self-supervised tasks using detected object traces. §5 shows results on object trace detection and representation learning for object detection. §6 concludes and suggests future directions.

## 3 Object Trace Detection

Our object trace detection algorithm involves three steps: (1) preprocessing, (2) object cluster proposal, and (3) object trace tracking. We elaborate each step below.

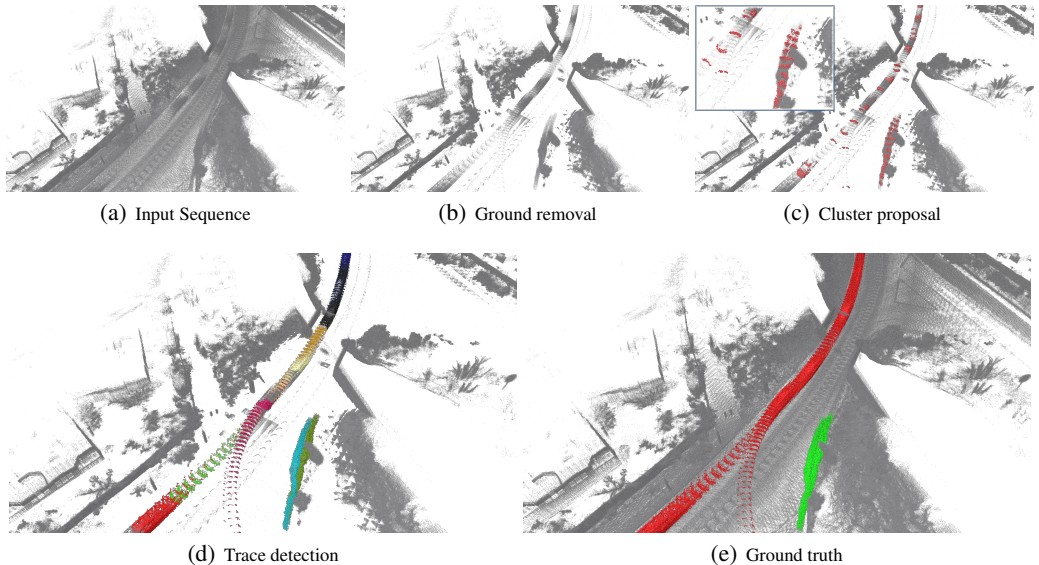

(a) Input Sequence      (b) Ground removal      (c) Cluster proposal

(d) Trace detection      (e) Ground truth

Figure 3: Object trace detection. In (a), we visualize the input point cloud sequence in world coordinates. Figures (b) to (d) show steps of trace detection. In (c), we highlight a subset of proposed object clusters in red. In (d), we color each detected trace differently. In (e), we visualize the ground truth, where vehicles are red and pedestrians are green.

**Sequence preprocessing.** We apply two preprocessing steps on each point cloud sequence before passing it to our detection algorithm. First, we bring all frames into the world coordinate system using the provided ego-motion (Figure 1, left). Second, we remove points on the ground, which typically account for 50% to 90% of the points; this improves detection efficiency and robustness. Appendix A provides preprocessing details.

**Object cluster proposal.** Since object sizes vary across classes, we use point cloud segmentation to extract point clusters corresponding to object instances. Unlike existing LiDAR point cloud segmentation methods, e.g. [13], we only extract segments corresponding to moving object instances.

Given a point cloud sequence, we use motion as a cue for segmentation. We estimate the velocity $\boldsymbol{v}_i \in \mathbb{R}^3$ of each point $\boldsymbol{p}_i$ and compute a pairwise proximity score $s_{ij}$ via

$$s_{ij} = \exp\left(-\frac{\|\boldsymbol{p}_i - \boldsymbol{p}_j\|^2}{\sigma_p^2} - \frac{\|\boldsymbol{v}_i - \boldsymbol{v}_j\|^2}{\sigma_v^2}\right), \tag{1}$$

where $\sigma_p = 0.5$ m and $\sigma_v = 10$ m/s. $\sigma_v$ is chosen to be large to reduce variance in velocity estimation. We then run normalized cuts [46] with edge weights $\{s_{ij}\}$ and select clusters with average velocity $> 0.1$ m/s and belong to the same frame.

To estimate motion, we adapt two state-of-the-art methods to point cloud sequences (details in §5). Our first variant "NeuralSF" follows Li et al. [47]. To accommodate point cloud sequences, we optimize for a spatiotemporal velocity field $f_\theta : \mathbb{R}^4 \to \mathbb{R}^3$ mapping any point location $\boldsymbol{x} \in \mathbb{R}^3$ and time $t$ to a velocity $\boldsymbol{v} \in \mathbb{R}^3$. We normalize $(\boldsymbol{x}, t)$ to $[0, 1]^4$ and optimize network parameters $\theta$ using chamfer distance between adjacent frames. Our second alternative "SSL-SceneFlow" follows Mittal et al. [48], who estimate scene flow between adjacent frames. They train using self-supervision on KITTI and nuScenes. We apply their pretrained model to each pair of adjacent frames.

**Object trace tracking.** Assuming each point cluster rigidly transforms across frames, we adopt a multi-object tracking algorithm using Kalman filtering to track cluster motion [49]. Specifically, we use a Kalman filter to track the velocity of each object's center from a sequence of centers. For each cluster, if tracking succeeds, the output of tracking is a trace (§2). Appendix B provides details.

## 4 Self-Supervised Tasks

Given a set of object traces from moving object instances in LiDAR sequences, we design self-supervised tasks that yield point cloud features for downstream tasks like single-frame object detection.

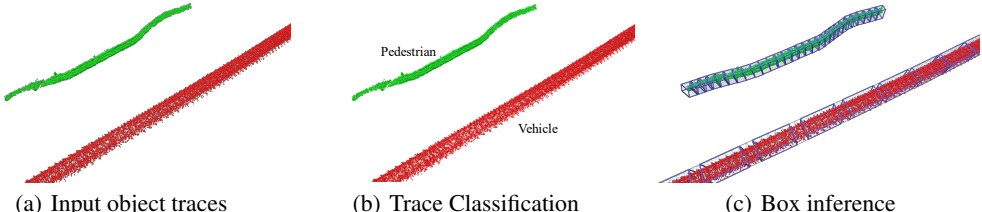

| (a) Input object traces | (b) Trace Classification | (c) Box inference |

Figure 4: Box inference. For visualization, (c) highlights the estimated bounding boxes.

Here we introduce two self-supervised tasks in §4.1 and §4.2, used to train a feature extractor with single-frame point cloud input. §4.3 provides details of our feature extraction module.

## 4.1 Task I: Motion Segmentation

We consider motion segmentation as the first self-supervised task. Since we extract traces with high precision, we can use them as segmentation masks to train a single-frame model differentiating moving objects from the background. We train a pointwise classifier that classifies each feature vector as either *moving* or *non-moving*, implemented as a multi-layer perceptron (MLP) combining ReLU [50] and BatchNorm [51] layers beyond the architecture in §4.3. Our loss is negative log-likelihood.

To generalize from moving to static objects, we augment each scene with randomly-sampled point clusters corresponding to moving objects from other scenes. In particular, for each scene, we randomly choose 50 point clusters and place them at a random locations.

## 4.2 Task II: Object Detection

Task I is straightforward but fails to use all available information. For example, object orientations can be inferred from motion but are ignored. Here, we estimate 3D bounding boxes and class labels from each object trace and use this information to train a 3D object detector; see Figure 4. We use the same box regression component and regression/classification loss as Yin et al. [52]. Estimating 3D bounding boxes to train the object detector from traces involves several subproblems:

- **Registration:** We estimate the velocity of each object cluster in each trace while enforcing smoothness. This enables us to approximately reconstruct the object by incorporating geometry collected from multiple frames.
- **Trace classification:** We categorize each object trace into one of the movable object classes. This is the only sub-problem that requires labeled data. We emphasize that class label is much cheaper to acquire compared to 3D bounding box annotations.
- **3D bounding box estimation:** Given estimated object class labels, we estimate the 3D bounding boxes for each object class. We propagate the box size estimate from densely reconstructed objects to sparse objects in each single frame, since the former provide high-confidence estimates.

The end result of the procedure above is a set of 3D bounding boxes that can be used to train a single-frame object detector. Details of these steps are provided below.

**Registration.** As observed in Table 1, most objects in the Waymo dataset follow smooth trajectories. For robustness against irregular sampling of LiDAR point clouds, we use this property to help estimate a velocity $\boldsymbol{v}_i \in \mathbb{R}^3$ for each object cluster $C_i$ of frame $i$. We optimize

$$\min_{\boldsymbol{v}_i} \sum_{\substack{(i,j): \\ |i-j|=1}} \left( \lambda \|\boldsymbol{v}_i - \boldsymbol{v}_j\|^2 + \sum_{\boldsymbol{p} \in C_i} \frac{1}{|C_i|} \min_{\boldsymbol{q} \in C_j} \|\boldsymbol{p} + \boldsymbol{v}_i - \boldsymbol{q}\|^2 \right), \tag{2}$$

where $\lambda = 1$. The estimated $\{\boldsymbol{v}_i\}$ brings all object points to the same coordinate system, forming a denser point cloud of the object instance, enabling trace classification and 3D box estimation.

**Trace classification.** Since object class labels are semantic rather than geometric, some minimal supervision is needed to distinguish classes (car, pedestrian, cyclist). We extract ground truth object class labels for 1,500 object instances (600 cars, 600 pedestrians and 300 cyclists) out of around 8 millions object instances (or 26,000 object traces) in our training dataset to train an object classifier

| | Vehicle | | | | Pedestrian | | | | Cyclist | | | |
|---|---|---|---|---|---|---|---|---|---|---|---|---|
| | IoU=0.6 | | IoU=0.7 | | IoU=0.4 | | IoU=0.5 | | IoU=0.4 | | IoU=0.5 | |
| | L1 AP | L2 AP | L1 AP | L2 AP | L1 AP | L2 AP | L1 AP | L2 AP | L1 AP | L2 AP | L1 AP | L2 AP |
| Ours-LP1 | 17.5 | 17.0 | 6.1 | 5.9 | 28.4 | 26.2 | 21.2 | 19.6 | 59.9 | 57.8 | 48.5 | 46.8 |
| Ours-LP2 | 43.7 | 39.6 | 15.1 | 13.8 | 45.6 | 40.1 | 31.6 | 27.2 | 63.7 | 61.2 | 49.7 | 47.3 |
| Ours-no-seg | 52.8 | 47.8 | 21.4 | 19.3 | 51.4 | 45.4 | 32.9 | 28.0 | 63.3 | 60.7 | 48.7 | 46.3 |
| Ours | 53.6 | 48.8 | 22.3 | 20.6 | 51.6 | 45.9 | 33.7 | 28.9 | 64.6 | 62.0 | 50.1 | 47.6 |

Table 2: Single-frame object detection results for unsupervised learning approaches. Here LP1/LP2 stands for the trained model before the first/second iteration of label propagation.

that takes an object trace and outputs an object class label. An additional object class "Other" is included to deal with outliers. We label 3,000 instances of outliers as "Other". Our object trace classifier is adapted from Point Transformer [53] and Voxel-based architectures, with 5 transition-down layers of dimension 32, 64, 128, 256, and 256, respectively. Each trace is represented as a 4D matrix with each row representing the location and time of a point.

**Bounding box estimation.** In LiDAR object detection, each 3D bounding box is represented by a 7D vector with location, size, and orientation. Since object sizes vary among classes, we group object traces based on the estimated object class labels. For each object class, we observe that there is a fraction of objects that are densely captured by LiDAR while the rest only represent object parts.

Therefore, we learn a model that regresses the bounding box size for each object trace, assuming it does not change along each trace. For each class, we select the top 20% of object traces ranked by number of points. We then use the velocity computed in (2) to bring point clusters into the same coordinate system and compute a 3D bounding box that covers all points with minimal volume. The resulting bounding box size vectors are used to train a model that regresses bounding box size. We use the model to predict bounding box sizes from all other object traces. The model architecture and input data representation of the regression model are also adopted similarly as in trace classification.

Given the box size, we solve an optimization problem to estimate box location and orientation:

$$\min_{\boldsymbol{b}_i} \sum_{(i,j):|i-j|=1} d_1(\boldsymbol{b}_i, \boldsymbol{b}_j) + d_2(\boldsymbol{b}_i, C_i). \tag{3}$$

Here, $\boldsymbol{b}_i$ represents box attributes, $d_1$ penalizes differences in orientation and enforces smoothness of box locations, and $d_2$ encourages the $i$-th box $\boldsymbol{b}_i$ to cover the $i$-th point cluster $C_i$. See appendix C.

**Moving-to-static label propagation.** The resulting 3D bounding boxes only represent moving objects in each sequence. To promote generalization, we notice that static clusters of movable objects can be extracted but were rejected in the object cluster tracking step (see section 3). We train a classifier on *registered* object traces and ask it to find out static clusters that are geometrically similar. In addition, we train a single-frame object detector using estimated 3D bounding boxes and apply it to predict 3D bounding boxes on the training data. Due to the geometric similarity between moving and static objects, the object detector generates 3D bounding boxes for static objects. To improve robustness, we then apply [49] to verify consistency of temporally-adjacent bounding boxes and reject false positives. The verified 3D bounding boxes for static objects are added to the training set to train another single-frame object detector from scratch. We repeat this procedure twice, yielding a single-frame object detector that reasonably generalizes to moving and static objects. Appendix D provides details and visualization.

### 4.3 Model Architecture

We modify the CenterPoint architecture [54, 52] to construct our single-frame feature extractor, which outputs a 256-dimensional feature for each 2D grid cell of size $0.1\text{m} \times 0.1\text{m}$ in the $x$-$y$ plane. The architecture is identical to the "VoxelNet" variant of CenterPoint that uses the SpConv Library for feature extraction [55]. The bounding box regression layer is modified according to self-supervised tasks discussed later. The training parameters are set to the same as the Waymo dataset defaults.

## 5 Experiments

We perform experiments related to each step of our proposed approach. Unlike most self-supervised learning works that require fine-tuning on labeled data, our self-supervised tasks are closer to the

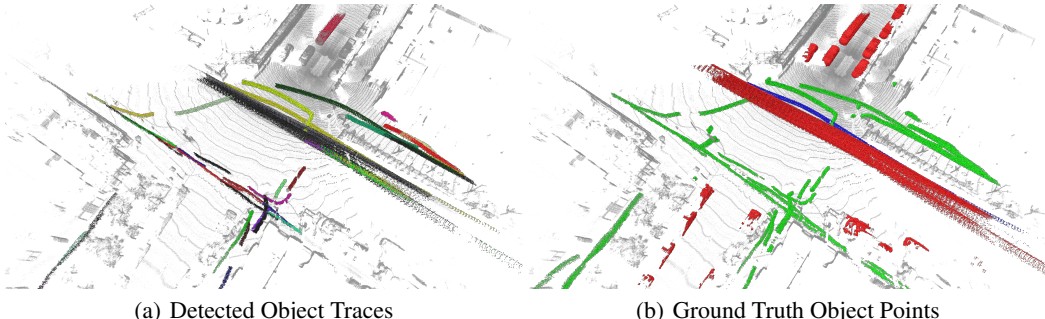

| (a) Detected Object Traces | (b) Ground Truth Object Points |

Figure 5: Qualitative results of our detected object traces. Left: Results of our object trace detection algorithm, each object trace is in different color. Right: Ground truth points corresponding to moving or static objects with different color (red: vehicle, blue: cyclist, green: pedestrian).

downstream tasks, enabling us to train a single-frame object detector with only a small set of class labels are needed but not any 3D box labels. We therefore perform two sets of evaluation on object detection algorithms, corresponding to the unsupervised setting and the self-supervised setting. In addition, we evaluate the quality of our detected object traces to demonstrate the reliability of our object trace detection algorithm.

§5.1 compares variants of our object trace detection algorithms in terms of the quality of the object traces. §5.2 compares our unsupervised single-frame object detector trained via self-supervised tasks described in §4 without any annotated 3D bounding boxes. §5.3 compares our single-frame object detector trained under different settings.

**Dataset.** We conduct our experiments on the Waymo Open Dataset for high-quality, dense point cloud sequences [1]. The dataset contains 798 training sequences and 202 validation sequences, with each sequence containing $\sim 200$ frames with a sampling frequency of 10Hz.

## 5.1 Object Trace Detection Results

Here, we evaluate the quality of detected object traces using the methods in §3. We compare two algorithmics, NeuralSF and SSL-SceneFlow. Both adapt state-of-the-art scene flow estimation algorithms to point cloud sequences. We use the Waymo Open Dataset training split for both.

**Evaluation protocol.** We report the mean Intersection over Union (mIoU) between the detected object traces and the ground truth moving objects. For each detected point cluster, we find the 3D bounding box closest to the geometric center of the cluster. We compute the IoU between points in the box and the ones in the cluster. We also use the class labels of the box to categorize

| Box count | Car | Pedestrian | Cyclist | Other |
|-----------|-----|------------|---------|-------|
| Neural-SF | 1027229 | 729275 | 29178 | 348452 |
| SSL-SF | 731480 | 631745 | 28369 | 543206 |
| GT | 1161019 | 1427211 | 42411 | - |
| *Mean IoU* | Car | Pedestrian | Cyclist | Other |
| Neural-SF | 70.3 | 95.5 | 96.5 | - |
| SSL-SF | 67.4 | 90.0 | 91.6 | - |

Table 3: Object trace detection results.

each cluster. If IoU is zero, we categorize the detected cluster as "Other." We use the class labels for evaluation purpose only and report the evaluation results in Table 3.

**Analysis.** Table 3 provides object trace detection results; Figure 5 qualitatively compares our detected object traces using Neural-SF to the ground truth object points. Both algorithms achieve reasonable performance finding object traces from point cloud sequences. Neural-SF outperforms SSL-SF, possibly due to the joint estimation of motion among all frames, which provides robustness against false positive and false negative motions and promotes smoothness of motion.

## 5.2 Unsupervised Object Detection Results

Here, we evaluate the performance of our single-frame object detector trained with minimal class labels. We use object traces from Neural-SF, due to their superior quality. Since multiple iterations of label propagation are involved (§4), we report the performance for both our final model and the

intermediate models after each label propagation iteration; we also try to remove motion segmentation (task I) and repeat the procedure for object detection (task II) only.

**Evaluation protocol.** We report L1 and L2 Average Precision (AP) on the validation split of the Waymo dataset for vehicles, pedestrians, and cyclists. We use two sets of IoU thresholds to compute AP for each object class: $\{0.6, 0.7\}$ for vehicles, $\{0.4, 0.5\}$ for pedestrians, and $\{0.4, 0.5\}$ for cyclists. One of the IoU thresholds for each class is identical to the standard choice. See Table 4.

**Analysis.** Our unsupervised single-frame object detector works well for all classes (see Table 2). Among all three object classes, we achieved the highest precision on cyclist class, which reflects the quality and coverage of our detected object traces for each object class reported §5.1. Empiricailly, we found that adding more iterations of label propagation do not improve overall performance. A possible reason is that the label propagation algorithm is sensitive to hyper-parameters.

### 5.3 Self-supervised Object Detection Results

In this section, we compare with two state-of-the-art self-supervised learning approaches: PointContrast [17] and DepthContrast [16]. These two methods work on single-frame point clouds but do not test on the Waymo Open dataset. We implement these two methods with a CenterPoint [52] pipeline. For ablation study, we train our model with three options: 1) (task-I) only task I's loss is used, 2) (task-II) only task II's loss is used, 3) (all) the sum of both tasks' loss is used.

Following the self-supervised learning routine, we split our training set into two parts and use all training point cloud sequences to pretrain a feature extraction module without using any labels. Then, we fine-tune the pretrained feature extraction module on 10% of the training point cloud sequences. The evaluation metrics follow §5.2; we use standard IoU thresholds for this experiment, i.e., 0.7, 0.5 and 0.5 for vehicle, pedestrian and cyclist, respectively.

**Analysis.** Table 4 summarizes our results. PointContrast and DepthContrast improve overall AP by $0.6\%$ and $0.2\%$, resp. Our self-supervised tasks achieved $0.8\%$, $1.9\%$ and $2.6\%$ improvement for task I (motion segmentation), task II (object detection), and all tasks, resp. Among all three classes, we achieved the highest improvement on cyclists due to quality of the detected object traces, agreeing with the results from §5.2 and §5.1.

|  | Vehicle | Pedestrian | Cyclist |
|---|---|---|---|
| (10% train) | 60.6 | 60.7 | 65.4 |
| PointContrast | 61.3 | 61.3 | 66.0 |
| DepthContrast | 60.9 | 60.7 | 65.7 |
| Ours-task-I | 61.5 | 61.4 | 66.3 |
| Ours-task-II | 62.2 | 62.6 | 67.6 |
| Ours-all | 63.3 | 63.0 | 68.4 |
| (100% train) | 66.5 | 67.7 | 69.3 |

Table 4: Single-frame object detection results for self-supervised learning.

## 6 Discussion and Conclusion

Our object trace detection algorithm reliably extracts objects that consistently and smoothly move in a point cloud sequence. These easily-extracted moving object traces provide critical information for downstream object detection without supervision, uncovered using our self-supervised tasks. Our design choices differ from other auto-labeling approaches, emphasizing quality over quantity of the pseudo-labels. Even though some class labels are needed to achieve state-of-the-art class-aware object detection performance, our work reveals the power and ease of obtaining high-precision/low-coverage pseudo-labels to train complex 3D shape analysis tasks like object detection.

**Limitations and Future Work.** Our method requires a minimal amount of categorical labels to distinguish between different objects; we view this requirement as sensible, in that the *type* of moving object (vehicle, pedestrian, cyclist) requires semantic knowledge. Moreover, the painstaking process of annotating object bounding boxes in typical 3D supervision is not needed in our setting: we just need a class label for our pre-detected object boxes (one of three discrete options, rather than a position of a box in 3D). If such supervision is removed, a candidate replacement of the learned object classifiers might be to cluster the moving objects based on their geometry alone; our attempts at this unsupervised approach decreased performance. Incorporating self-supervised or unsupervised object classifier models might be a promising direction to better automate this pipeline.

Our method focuses on movable objects like pedestrians, cyclists, and vehicles, since other classes will not be captured by object trace detection. Different pretext tasks are needed for detecting static object classes. One could consider spatio-temporal structures and complementary views.

**Acknowledgments**

The MIT Geometric Data Processing group acknowledges the generous support of Army Research Office grants W911NF2010168 and W911NF2110293, of Air Force Office of Scientific Research award FA9550-19-1-031, of National Science Foundation grants IIS-1838071 and CHS-1955697, from the CSAIL Systems that Learn program, from the MIT–IBM Watson AI Laboratory, from the Toyota–CSAIL Joint Research Center, and from a gift from Adobe Systems. The Toyota Research Institute provided funds to support this work.

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
