# OpenReview forum: "Representation Learning for Object Detection from Unlabeled Point Cloud Sequences"
_robot-learning.org/CoRL/2022/Conference — CoRL 2022 Poster_

### Official Review · Reviewer_Thig · 2022-07-20

**Originality:** Good
**Technical Quality:** Good
**Clarity Of Presentation:** Good
**Impact:** 3

**Recommendation:**

Weak Accept: I recommend accepting the paper, but will not argue for my recommendation if the majority of other reviewers have a different opinion.

**Summary:**

This paper aims at reducing the effort of point cloud labeling by learning feature representations from point cloud sequences. The authors propose a method to extract moving object traces from a sequence of pose-aligned points clouds. The traces are computed by removing the ground plane and tracking point clusters. The points are clustered based on spatial proximity and scene flow between all point pairs in the sequence. The object traces are used to pre-train a single-scan feature extractor on tasks like motion segmentation and object detection without the need for 3D bounding box labels. Finally, the experimental evaluation shows the effectiveness of the object trace detection as well as the performance improvement by the proposed pre-training strategy.

**Issues:**

My main concern is regarding the use of the term "self-supervised". The authors claim to "propose a technique for representation learning from unlabeled LiDAR point cloud sequences". They mention that a "minimal amount of categorical labels" is still needed. In my view, the claim of performing "self-supervised tasks" does not hold since in fact labels are used, even though they are easier to get than bounding box annotations. Also, it is not clear if the ground truth poses are used for pre-processing which one could see as another form of labeling if manual inspection and validation of poses are carried out to obtain a good ground truth odometry. In general, I recommend updating the terminology and not using the term "self-supervised tasks" which is too strong in my view.

Minor issues:
- It is very hard to see details in Fig. 1 on the right or in Figure 3 at the top. Is it possible to zoom in to better focus on details than the complete scene?
- What is the color coding in Figure 5 on the right? Using the same color makes it hard to distinguish individual traces here.
- What extractly are L1 and L2 AP? Is this a metric coming from the Waymo Open Dataset? Please specify.
- The section linking is broken (always jumps to the first page).
- Keyword "object detectioni" is misspelled.
- When mentioning the d1 and d2 losses in Eqn. (3), please also cite [3] as done in the supplementary material.

**Quality Of The Limitations Section:**

Limitations are addressed clearly

**Reviewer Expertise:**

4: The reviewer is confident but not absolutely certain that the evaluation is correct

**Robotics Focus:**

Relevant but unlikely to deploy to hardware in near future

**Strengths And Weaknesses:**

Strengths:
- The idea of focusing on easy-to-extract high-quality labels is interesting and the experimental evaluation shows that these labels can be used to improve object detection.
- Two scene flow methods in the experiments are trained on KITTI and nuScenes data and still deliver suitable results on the Waymo data. This experiment validates their use for extracting object traces based on position and velocity similarity.
- The paper is in general well written and easy to follow.

Weaknesses:
- The method relies on a "provided ego-motion". It is unclear how the performance is affected when odometry has to be estimated online possibly resulting in more drift.
- The experimental analysis in Section 5.2 is incomplete since it only covers the performance for different classes. I recommend commenting on the effect of the different pre-training stages.
- The first downstream task is a single-frame motion segmentation. However, I doubt that this task is meaningful since motion can not be reliably estimated from a single scan.
- The object cluster proposal computes pairwise proximity scores between all points in the sequence. Since sequences of 200 frames result in a lot of point correspondences, I wonder how this affects the runtime of the object trace detection.

**Summary Of Recommendation:**

The paper provides an interesting approach to using temporal LiDAR data to extract high-quality object traces used for representation learning of single-scan downstream tasks. The experiments show the effectiveness of the trained object detector and that the detection performance can be increased with fine-tuning compared to existing representation learning approaches. However, I still see some weaknesses and issues mainly affecting the contribution of self-supervision and advice resolving the unclarities.

---

### Official Review · Reviewer_442W · 2022-07-24

**Originality:** Very Good
**Technical Quality:** Good
**Clarity Of Presentation:** Good
**Impact:** 4

**Recommendation:**

Weak Accept: I recommend accepting the paper, but will not argue for my recommendation if the majority of other reviewers have a different opinion.

**Summary:**

The paper proposed a pretraining method for 3D object detection in point clouds. Leveraging the fact that a large portion of objects are moving smoothly in a sequence of frames in the Waymo dataset, and that moving objects leaving a trace in a sequence of point clouds are relatively easy to extract without manual labeling, pseudo 3D labels of objects can be automatically extracted, and the network can be pretrained on the detection task and related tasks. Experiments are conducted on the Waymo dataset and achieved better performance than existing pretraining methods.

**Issues:**

1. Explanation of how tasks 1 and 2 work together.
2. Reference to table 2 in the main text.
3. Explanation of how the problem of low proportion of moving cars is addressed.
4. Discussion on the claim that using high-precision / low coverage pseudo labels is beneficial.

**Quality Of The Limitations Section:**

Limitations are addressed clearly

**Reviewer Expertise:**

4: The reviewer is confident but not absolutely certain that the evaluation is correct

**Robotics Focus:**

Highly relevant to robotics but no hardware experiments

**Strengths And Weaknesses:**

Strengths:
1. Proposed to leverage the object traces as a clue for automatic labeling for the pretraining of the object detection task, which is an insightful observation and inspiring.
2. Designed a series of steps to extract object traces and estimate the velocity and 3D bounding boxes of the objects from them. The steps are overall explained well and have practical value.
3. Experimental results are reported on each step of the proposed method and support the claimed improvement of the paper.

Weaknesses:
1. The description of the network training is somewhat vague. For example, It is not clear how the task 1 and 2 are combined in the method. Are they two heads in the network and trained together? Or are they trained sequentially? Accordingly it is not clear how the networks in table 2 are trained (whether each task is included and whether they are iterated).
2. Table 2 is not referenced in the paper. It is hard to find the corresponding paragraph in the main text.
3. From table 1, the moving cars only account for 1/4 of all cars in the dataset. How is this problem addressed? Is it through the moving-to-static label propagation by adding newly detected objects to the pseudo labels in two iterations of training? Does adding a third iteration help? More discussion in section 5.2 could help, which also better supports the claim of benefit using high-precision/low-coverage pseudo labels. The latter is an interesting claim but it is not discussed much in the main text.


**Summary Of Recommendation:**

The paper proposed a practical pretraining method for point cloud object detection task. The approach of leveraging object traces is inspiring. The experimental results are generally convincing. Some minor revision on the organization and additional explanation and discussion are desired before it qualifies for publication. Therefore I suggest weak acceptance.

---

### Official Review · Reviewer_NRFL · 2022-07-30

**Originality:** Very Good
**Technical Quality:** Good
**Clarity Of Presentation:** Fair
**Impact:** 4

**Recommendation:**

Weak Accept: I recommend accepting the paper, but will not argue for my recommendation if the majority of other reviewers have a different opinion.

**Summary:**

This paper proposes an unsupervised learning pipeline to detect dynamic objects from a sequence of point cloud measurements. The proposed pipeline consists of object traces detection, motion segmentation, and single-frame object detection. The single-frame object detection method has been tested on Waymo Open Dataset and provides improved performance.

**Issues:**

- Since the object trace detection accuracy limits the object detection performance, how is the performance affected by the object's motion? For example, the moving velocity, sudden movements, and turning angle.
- Kalman filter is mentioned on page 3, but it is unclear how it is included as it is never discussed later.
- In keywords: "detection" in misspelled as "detection".
- What are "pretext tasks"?

**Quality Of The Limitations Section:**

Limitations are addressed clearly

**Reviewer Expertise:**

4: The reviewer is confident but not absolutely certain that the evaluation is correct

**Robotics Focus:**

Highly relevant to robotics but no hardware experiments

**Strengths And Weaknesses:**

**Strengths**
- The proposed pipeline that utilizes point cloud sequence for dynamic object detection is novel. This design enables dynamic object detection with unlabeled data.
- The overall proposed pipeline is reasonable.
- The experiments are performed on real-world data.

**Weakness:**
- The proposed method consists of multiple modules that can perform multiple tasks—resulting in a heavy paper that includes all aspects of object trace detection, motion segmentation, and object detection.
- This reviewer finds the methodology part of the paper unclear. In particular, I am not confident in replicating the methods as not all steps are communicated. I recommend adding one or more algorithmic implementations of the proposed automatic labeling (main contribution) with precise explanations of inputs, algorithms, and outputs.
-  Focusing more on one aspect, e.g., object detection and the advantage of self-supervised learning, might be better for a conference paper. For example, the comparison between two object trace detection methods (NeuralSF and SSL-SceneFlow) can be removed or moved to the appendix since these are two published methods.
- A sequence of measurements is required to perform "single-frame" object detection. Thus it increases the required amount of data. In addition, the performance of object detection is limited by the performance of object trace detection.
- Self-supervision is great. But it is also better to report some supervised methods to see the existing gap.



**Summary Of Recommendation:**

This paper proposes a framework for dynamic object detection by detecting object traces using a sequence of point clouds. The paper contains a good idea on automatic labeling of LIDAR data. But it requires a major revision and improved clarity. See my comments in the Weaknesses section.

---

### Official Review · Reviewer_FBsS · 2022-08-04

**Originality:** Good
**Technical Quality:** Very Good
**Clarity Of Presentation:** Very Good
**Impact:** 4

**Recommendation:**

Weak Accept: I recommend accepting the paper, but will not argue for my recommendation if the majority of other reviewers have a different opinion.

**Summary:**

This paper proposes an approach for detecting moving objects from unlabelled point cloud sequences, without any annotations. The detected object traces are then used to produce point cloud features for downstream tasks like single-frame object detection using self-supervised tasks. The experiments are run on the Waymo Open Dataset.



**Issues:**

Issues
1. Can the approach be adapted to handle non-smooth trajectories? Or is this a core, critical assumption?
2. What pretext tasks could be constructed to extract representations for stationary objects?
3. Are the produced bounding boxes (after the object-detection pretext task) sufficient for reliable control? How much is performance affected, if at all ? (As compared to using labelled data).



**Quality Of The Limitations Section:**

Limitations are addressed clearly

**Reviewer Expertise:**

3: The reviewer is fairly confident that the evaluation is correct

**Robotics Focus:**

Highly relevant to robotics but no hardware experiments

**Strengths And Weaknesses:**

Strengths

1. Manual annotation is a major bottleneck for scaling up the amount of data that can be used for learning. This paper uses the idea of using motion to separate out moving objects. This can be used on any unlabelled point cloud sequence, which can be easily collected without the need for expensive bounding box annotation. Further, the approach includes self-supervised pretext tasks to use these traces to perform useful downstream tasks.



2. The paper includes thorough experimental evaluation that evaluate the quality of object traces, as well as the point cloud features for downstream tasks, showing that the proposed approach can produce reasonable traces and object detection results as compared to prior approaches. The authors also include qualitative visualizations that show the structure of the detected traces.


Weaknesses

1. The first step of the approach is extracting object traces from multi-timestep data. The paper assumes that objects move along a smooth trajectory in order to extract this. This might be a reasonable assumption for the waymo dataset included in the experiments, however it limits the general applicability of the approach to other settings, such as a manipulation robot interacting with objects to learn skills, which often have non-smooth behavior. The smoothness assumption is also used in the self-supervised object-detection task which is used to produce 3D bounding boxes from the object traces.

2. Central to the approach is extracting objects based on motion. There might be some objects of interest however that do not move, that we might still want to model. Though the paper does include moving to static label propagation, where 3D bounding boxes generalize due to visual similarity, this might be a special case of the dataset considered, and not hold in general for other domains.

3. While the paper shows evidence for the efficacy of the approach in obtaining useful bounding boxes for key object categories, the paper doesn’t include any experiments showing that these are sufficient for control. Evidence that any loss in the quality of produced bounding boxes does not lead to major loss in downstream control performance would increase the community’s confidence in adopting this approach.


**Summary Of Recommendation:**

I recommend acceptance, since using unlabelled point cloud sequences has the potential to scale up the amount of data used for learning, leading to much better performance for systems. The proposed idea relies on identifying objects based on motion, and while this has some issues as discussed, it can be a powerful source of self-supervision.

---

### Meta-Review · Area_Chair_ox57 · 2022-08-04

**Recommendation:** Accept (Poster)
**Confidence:** 5

**Metareview:**

The paper introduces a new method for learning feature representations from point cloud sequences and applying the learned feature representation to object detection in point clouds. The method automatically segments moving objects from point cloud sequences and uses object trace detection and object detection as self-supervision for feature learning.

The reviewers acknowledge that the idea of using moving object segmentation and self-supervision to learn feature representations of point clouds is novel. The conducted experiments support the proposed method.

There are concerns on the limitation of the proposed method on detecting static objects. Certain parts of the paper are not clear to the reviewers such as parts in the methodology and the network training. There is one concern that the proposed method still uses a small amount of labeled data which is not fully self-supervised.

After the rebuttal, the reviewers agree on the contribution and novelty of the paper. These are still concerns on the self-supervision claim in the paper, and the authors can revise the final paper according to the reviewers.


**Best Paper Nomination:**

No